# Aromatic Characterisation of *Moscato Giallo* by GC-MS/MS and Validation of Stable Isotopic Ratio Analysis of the Major Volatile Compounds

**DOI:** 10.3390/biom14060710

**Published:** 2024-06-16

**Authors:** Mauro Paolini, Alberto Roncone, Lorenzo Cucinotta, Danilo Sciarrone, Luigi Mondello, Federica Camin, Sergio Moser, Roberto Larcher, Luana Bontempo

**Affiliations:** 1Fondazione Edmund Mach, Via Mach 1, 38098 San Michele all’Adige, Italy; alberto.roncone@fmach.it (A.R.); lorenzo.cucinotta@unime.it (L.C.); f.camin@iaea.org (F.C.); sergio.moser@fmach.it (S.M.); roberto.larcher@fmach.it (R.L.); luana.bontempo@fmach.it (L.B.); 2Messina Institute of Technology, Department of Chemical, Biological, Pharmaceutical and Environmental Sciences, Former Veterinary School, University of Messina, Viale G. Palatucci snc, 98168 Messina, Italy; danilo.sciarrone@unime.it (D.S.); luigi.mondello@unime.it (L.M.); 3Chromaleont s.r.l., Messina Institute of Technology, Department of Chemical, Biological, Pharmaceutical and Environmental Sciences, Former Veterinary School, University of Messina, Viale G. Palatucci snc, 98168 Messina, Italy; 4Center Agriculture Food Environment (C3A), University of Trento, Via Mach 1, 38010 San Michele all’Adige, Italy

**Keywords:** *Moscato Giallo*, volatile compounds, GC-MS/MS, GC-C/Py-IRMS

## Abstract

Among the *Moscato* grapes, *Moscato Giallo* is a winegrape variety characterised by a high content of free and glycosylated monoterpenoids, which gives wines very intense notes of ripe fruit and flowers. The aromatic bouquet of *Moscato Giallo* is strongly influenced by the high concentration of linalool, geraniol, linalool oxides, limonene, α-terpineol, citronellol, hotrienol, diendiols, *trans/cis*-8-hydroxy linalool, geranic acid and myrcene, that give citrus, rose, and peach notes. Except for quali-quantitative analysis, no investigations regarding the isotopic values of the target volatile compounds in grapes and wines are documented in the literature. Nevertheless, the analysis of the stable isotope ratio represents a modern and powerful tool used by the laboratories responsible for official consumer protection, for food quality and genuineness assessment. To this aim, the aromatic compounds extracted from grapes and wine were analysed both by GC-MS/MS, to define the aroma profiles, and by GC-C/Py-IRMS, for a preliminary isotope compound-specific investigation. Seventeen samples of *Moscato Giallo* grapes were collected during the harvest season in 2021 from two Italian regions renowned for the cultivation of this aromatic variety, Trentino Alto Adige and Veneto, and the corresponding wines were produced at micro-winery scale. The GC-MS/MS analysis confirmed the presence of the typical terpenoids both in glycosylated and free forms, responsible for the characteristic aroma of the *Moscato Giallo* variety, while the compound-specific isotope ratio analysis allowed us to determine the carbon (δ^13^C) and hydrogen (δ^2^H) isotopic signatures of the major volatile compounds for the first time.

## 1. Introduction

Muscat has become a cultural phenomenon these days. Over the past few years, the demand for Muscat wine (also known as *Moscato* wine) has been on the rise, with drinkers searching for a sweeter, lighter wine with a low alcohol percentage. The global *Moscato* wine market is worth USD 1704.7 million, and it is expected to grow at an annual rate of 11.3% until 2030 due to the increasing demand for sweeter and low alcoholic grade wines [1]. The wine matrix is complex, comprising numerous components that play an important role in the perception of wine aroma and flavour as interactions among these compounds may occur through different mechanisms, resulting in the alteration of the chemical and sensory properties of wine [2,3]. Adulteration of wine is a worldwide issue affecting consumers and honest producers, with an estimated cost to the EU wine sector of around Euro 1.3 billion per year, 3% of the total sales value (JRC report). More than one million litres of counterfeit alcoholic beverages were seized across Europe in 2020 and over 1.7 million in 2021—the largest quantity being wine—in targeted actions regularly led by the European Anti-Fraud Office (OLAF) as part of joint Europol–Interpol operations called OPSON. 

For detecting fraudulent products, various analytical techniques have been developed and tested. Since the mid-1980s, isotopic analyses have also been integrated into the arsenal of techniques deployed to combat wine adulteration. Stable isotope ratios of biologically relevant elements can provide very important information about the geographical and botanical origin of food products. Therefore, they can be used to detect adulteration and fraud. When considering the carbon isotopic ratio in matrices of plant origin, most of the variability in these values is attributable to plant metabolism. There are three major groups of plants (C3, C4, CAM) that differ based on their photosynthetic cycle, with each group exhibiting characteristic ranges of isotopic values [4]. Additionally, there is some intra-variability within these ranges due to the climatic characteristics of the place of origin [5]. These differences can be exploited in various ways: tracing the botanical origin of a plant-derived product [6]; recognising the dietary preferences of an animal [7]; identifying adulterations in food products [8]; or reconstructing migratory routes of animals [9]. Oxygen and hydrogen isotopic ratios are intimately related to the water taken up by the plant from which a product is derived. The values of oxygen and hydrogen vary across the Earth’s surface depending on the geo-climatic characteristics of the areas traversed (humidity, altitude, distance from the sea, etc.). This means that plants growing in different parts of the globe will exhibit different hydrogen and oxygen isotopic values [10]. This property can, therefore, be used to trace the geographical origin of a certain product. Typically, this correlation is established with reference to samples of known origin, as is practiced with the official European isotopic wine database. 

In 1986, the European Union and then the International Organisation of Vine and Wine (OIV) officially adopted the stable isotope ratio analysis for the detection of different types of fraud (i.e., chaptalisation, watering, geographical and harvest year mislabelling). In countries like Italy, adding exogenous sugar to grape must is forbidden, leading to the development of specific techniques to detect such fraud. The official method OIVMA-AS-312-06 [11] involves measuring the δ^13^C on ethanol derived from wine distillation, where the presence of cane sugars can be detected. Since sugar cane is a C4 plant, its sugars have carbon isotopic values consistent with that. When added to grape must (from a C3 plant), the resulting alcohol will show shifted carbon isotopic values. However, the addition of beet sugar cannot be detected through δ^13^C analysis due to beet being a C3 plant. Instead, it is detected by the (D/H)_I_ ratio on the ethanol molecule after distillation (OIV MA-AS-311-05) [11,12,13]. Watering fraud can be revealed by the δ^18^O ratio of wine water after equilibration with a reference gas (CO_2_) (OIV MA-AS2-12) [11]. Vegetal water used for dilution is enriched compared to tap water, and by comparing with a database, watering can be detected. The EU official isotope wine databank (Regulation (EU) 2018/274) [14], including reference data, allows the definition of limits for authentic wines and musts in terms of isotopic data, tailored for each country, sub-area, and protected designation of origin (DOC-IGT) [15].

*Moscato* grape is broadly recognised as one of the most aromatic varieties and its scent is mainly due to the monoterpenes (terpene alcohols or terpenols), which give the wine its floral characteristic aroma. Monoterpenes are synthesised from glucose by acetyl-coenzyme A and are present in grapes in two forms: free and glycosidically-bound [16]. These compounds are mainly located in the peel of berries, but in aromatic varieties, they can also be found in pulp and juice [17]. 

Free forms are directly involved in the floral bouquet of grapes, whereas bound forms are non-volatile compounds that do not directly contribute to the aroma. These latter are constituted by an aglycone conjugated to a glycosyl group, like glucose, arabinose, ramnose or apiose [17,18]. Aglycones are mostly terpenols and terpenic polyols but can also be C_6_-alcohols, C_13_-norisoprenoids, benzenic compounds and others [19].

The relative distribution between the two forms depends on several factors, such as the ripening stage of berries or breeding [19,20]. Moreover, the bound glycosides forms can be transformed into the free counterparts through acidic or enzymatic hydrolysis [21]. Despite the characterisation of the Volatile Organic Compounds (VOCs) composition of *Moscato* wines being well known, few articles have considered their isotopic ratios. Spangenberg et al. [22] correlated the δ^13^C_(VOCs)_ of Pinot Noir wine with the predawn leaf water potential, observing that δ^13^C_(VOCs)_ are generally higher with an increasing vine water deficit. The authors noted that this approach is a useful tool to assess the changes in the water status of grapevine cultivars in different terroirs. Jin et al. [23] developed a SPME-GC-IRMS method for the analysis of the carbon stable isotope of six typical VOCs of wine (isoamyl acetate, 2-octanone, limonene, 2-phenylethanol, ethyl octanoate and ethyl decanoate) that could be used for authenticity assessment, which is less time consuming than the official method based on the determination of δ^13^C of ethanol after wine distillation. VOCs are among the most important molecules heavily/extensively contributing to the flavour and aroma of the wine. The synthesis of these compounds largely depends on environmental and biological factors, making them possible markers for wine authentication [23]. The authors concluded this preliminary research, stating that the efficiency of the method could be improved by adding other VOCs isotope ratios and by quantifying their relative concentration.

Extending the stable isotope ratio analysis to a larger number of volatile compounds may be more effective in checking for false declarations of *Moscato* wine. This paper aims to develop and validate robust analytical methods for the characterisation of volatile organic compounds in a particular type of *Moscato* wine (*Moscato Giallo*) or similar aromatic wines and for the determination of their compound-specific stable isotope ratios of carbon and hydrogen. For the first time to our knowledge, we provide here a complete characterisation of the δ^13^C and δ^2^H ratios (GC-C/Py-IRMS) and the relative quantification (GC-MS/MS) of the main aromatic terpenes in the grape must and in their corresponding wines from *Moscato Giallo* variety. 

## 2. Material and Methods

### 2.1. Samples Collection 

Seventeen different *Vitis vinifera* L. cv. *Moscato Giallo* grape varieties were collected from the Trentino Alto Adige and Veneto regions during the months of September and October 2021, from local farmers. For each variety, 200 grape berries were collected with pedicles from bunches and a portion of the samples was stored at −20 °C while the rest was subjected to alcoholic fermentation.

### 2.2. Winemaking

The grape berries were crushed, and the juice was separated from the pomace with a colander. Microvinification occurred in 1-L glass flasks, inoculating 500 mL of grape juice with a laboratory yeast strain (*Saccharomyces cerevisiae*, W303). Alcoholic fermentation was performed at 25 °C. The fermentation time course was monitored by determining the CO_2_ production, expressed as weight loss until the weight was constant. The end of the fermentation was confirmed by the measurement of glucose and fructose concentrations using Fourier transform infrared spectroscopy (FT-IR; WineScan™ FT 120; Foss, Hillerød, Denmark). Finally, the wine was then recovered by removing yeast cells and sediments by centrifugation.

### 2.3. Base Compositional Parameters of Must and Wine

The must and wine samples were analysed for basic parameters of maturity and quality. The main composition of musts (°Brix, sugars, pH, titratable acidity, density, tartaric acid, malic acid, potassium and readily assimilable nitrogen) and wines (alcoholic strength by volume, sugars, non-reducing extract, pH, total acidity, volatile acidity, density, tartaric acid, malic acid, lactic acid and potassium) were determined using FT-IR, previously calibrated according to the OIV official methods [11]

### 2.4. Sample Preparation and Extraction

Sample preparation and extraction were performed according to the method described by Paolini et al. [24]. The following were added to 50 g of the frozen berry sample: 0.5 g of gluconolactone; 100 µL of internal standard n-heptanol (230 mg/L); and 100 µL of nonyl-β-d-glucopyranoside (1000 mg/L). Homogenisation with Ultra-Turrax was performed at 21,000 rpm for 3 min and followed by centrifugation at 5,000 rpm for 10 min. The supernatant was recovered, and the solution was brought to an exact volume of 100 mL with ultrapure water (Arium Pro Lab, Sartorius AG; Göttingen, Germany). 

For the wine, 50 mL of the sample was diluted to 100 mL with Milli-Q water after the addition of the internal standard (100 µL of n-heptanol and 100 µL of nonyl-β-d-glucopyranoside).

Solid phase extraction was performed using ENV+ cartridges, 1 g (Biotage, Uppsala, Sweden). The cartridge was activated with 20 mL of methanol and 25 mL of Milli-Q water, and the sample was loaded onto the cartridge. Free VOCs were eluted with 30 mL of dichloromethane in a 100 mL glass boiling flask and dried with anhydrous sodium sulphate. Glycosidic VOCs were eluted with 30 mL of methanol, and this solution was first dried using a rotavapor and then dissolved in 4.5 mL of citrate buffer at pH 5. After the addition of 200 µL of a glycosidic enzyme with strong glycosidase activity (AR 2000 at 70 mg/mL in water), the solution was kept in a bath at 40 °C overnight to allow the release of the volatile compounds from the glycosidic bond. After that, 100 µL of n-heptanol was added, and the VOCs were extracted with SPE cartridges, as reported for free VOCs. 

After the GC-MS/MS analysis of the free and bound VOCs, the two fractions were merged and concentrated to obtain a proper signal for IRMS detection. For the concentration step, a double volume of pentane was added to the total extracted fraction to obtain a low boiling azeotropic mixture (dichloromethane/pentane, 1:2). The concentration was thus performed in a bath at 40 °C until 0.5 mL volume using a Vigreux distillation apparatus. 

### 2.5. GC-MS/MS Analysis

According to Paolini et al. [24], VOCs were analysed by using an Agilent Intuvo 9000 GC system coupled with an Agilent 7000 Series Triple Quadrupole MS (Agilent Technologies, Wilmington, DE, USA). The spectrometer was equipped with an electron ionisation source operating at 70 eV and the filament current was 50 μA.

Separation was obtained by injecting 2 μL in split mode (1:5) into a DB-Wax Ultra Inert (20 m × 0.18 mm id × 0.18 μm film thickness) capillary column with a constant He flow of 0.8 mL/min. The injector temperature was set at 250 °C. The oven temperature was programmed starting at 40 °C for 2 min, raised to 55 °C by 10 °C/min, then raised to 165 °C by 20 °C/min, and finally raised to 240 °C by 40 °C/min and held at this temperature for 5 min.

The mass spectra were acquired in the dynamic multiple reaction monitoring (dMRM) mode using N_2_ as collision gas (flow of 1.5 mL/min) and in the full scan mode (mass range from 33 to 400 *m*/*z*). The transfer line and source temperature were set at 250 °C and 230 °C, respectively.

Data acquisition and analysis were performed using the Agilent Technologies MassHunter Workstation software—Data Acquisition (ver. B.07.06) and the Agilent MassHunter Workstation Software—Quantitative Analysis (ver. B.08.00), respectively.

### 2.6. GC-C/Py-IRMS Analysis

Carbon and hydrogen stable isotope ratio analysis of monoterpenes in the grape and wine samples was carried out using a Trace GC Ultra fitted with a TriPlus autosampler (Thermo Fisher Scientific, Waltham, MA, USA), interfaced with an Isolink-IRMS (Delta V Advantage, Thermo Fisher Scientific) and connected in parallel with a single-quadrupole GC–MS (ISQ Thermo Scientific, Milan, Italy) for compounds identification. The stable isotope ratios of the monoterpenes in grapes and wines were determined in accordance with Khatri et al. [25]. As an internal standard, 2-octanol was added to the grape/wine extracts to normalise any variations during the analysis. A volume of 0.8 µL was injected at 250 °C in the splitless mode in a ZB-WAX (30 m × 0.32 mm × 0.5 µm, Phenomenex, Torrance, CA, USA) capillary column with He as the carrier gas, with the flow rate set at 2.3 mL/min.

The GC-oven temperature was initially kept at 40 °C for 3 min and subsequently set to increase by 3 °C/min to 55 °C, then by 5 °C/min to 165 °C, and by 10 °C/min until the final temperature of 240 °C was reached and maintained for 7.45 min. For the carbon isotope ratio analysis, the temperature of the combustion oven was set at 1000 °C, whereas for the hydrogen isotopic ratio, the pyrolyzer (HTC oven) temperature was set at 1400 °C.

All samples were analysed in triplicate, and the peak integration was carried out with the Isodat 3.0 software (Thermo Fisher Scientific).

The working standards (geraniol and linalool) were introduced every six injections to check for any possible drifts. The isotope values were presented as delta-notation relative to the international standards VPDB (Vienna-Pee Dee Belemnite) for δ^13^C and VSMOW-SLAP (Vienna Standard Mean Ocean Water-Standard Light Antarctic Precipitation) for δ^2^H, according to the Equation (1).
(1)$$\delta^{i}E_{sample/standard}=\frac{R{{(}^{i}E{/}^{j}E)}_{sample}}{R{{(}^{i}E{/}^{j}E)}_{standard}}-1$$
where *standard* is the international measurement standard, *sample* is the analysed sample and *^i^E/^j^E* is the isotope ratio between the heavier and lighter isotope. The delta values are multiplied by 1000 and expressed commonly in units “per mil” (‰) or, according to the International System of Units (SI), in unit ‘milliurey’ (mUr). The isotopic values were calculated against two standards through the creation of a linear equation.

The standards that have been used in the isotopic analyses were international reference materials or in-house working standards that have been calibrated against international reference materials, δ^13^C values against fuel oil NBS-22 (δ^13^C = −30.03‰) and sucrose IAEA-CH-6 (δ^13^C = −10.45‰) and δ^2^H values against Sicilian olive oil USGS 84 (δ^2^H = −140.4‰) and Ice-core water USGS 49 (δ^2^H = −397‰).

## 3. Results

### 3.1. Grape and Wine Composition

The composition of *Moscato Giallo* grape juice samples is reported in Table 1. Specifically, the total soluble solids (°Brix) ranged from 20.03 to 23.55 (median of 21.19), with a concentration of the reducing sugars between 152 and 225 g/L (median of 189 g/L). The pH ranged from 3.08 to 3.55 (median of 3.26), whereas the titratable acidity was between 4.68 and 7.74 g/L (median of 6.40 g/L). The concentration of organic acids was 3.65 (min)–5.47 (max) with a median of 4.79 g/L and 2.46 (min)–5.51 (max) with a median of 4.20 g/L, respectively, for tartaric acid and malic acid. The potassium (K) was quantified between 1.00 and 2.06 g/L (median of 1.39 g/L), whereas the YAN (yeast assimilable nitrogen taken as the sum of NH_4_^+^, free α-amino acids and some small peptides) was between the limit of quantitation (<20) and 163 mg/L (median of 83 g/L).

Based on the scientific literature, the above parameters fall into the technological variability characteristics of *Moscato* grape varieties, regardless of cultivation site, production year and climatic conditions [20,26,27].

The chemical–physical parameters used for the quality control of the resulting wines are reported in Table 2. As is shown, the reducing sugars were depleted in all fermentation trials (<1.00 g/L), leading to an ethanol content (EC) ranging from 11.03 to 12.28% *v*/*v* (median of 11.64% *v*/*v*). The non-reducing extract (nRE), known as the “body” in wine-tasting, was between 22.10 and 25.52 g/L (median of 24.07 g/L), whereas the total acidity (TA) ranged from 5.97 to 8.56 g/L (median of 7.41 g/L). These values satisfy the minimum requirements of the *Moscato Giallo* wine specified in the DOC Wines Production Disciplinary for the Province of Trento (EC ≥ 11.00% *v*/*v*, nRE ≥ 17.0 g/L, TA ≥ 4.5 g/L) [28] and Bolzano (EC ≥ 11.00% *v*/*v*, nRE ≥ 16.0 g/L, TA ≥ 4.0 g/L) and for the Veneto Region (EC ≥ 10.50% *v*/*v*, nRE ≥ 16.0 g/L, TA ≥ 5.0 g/L) [29].

Concerning the other parameters, the pH ranged from 3.08 to 3.46 (median of 3.26), the volatile acidity was under the limit of quantitation in all wines (<0.10 g/L), and the tartaric acid was between 2.16 and 0.94 g/L (median of 1.66 g/L). The malic acid was quantified between 3.07 and 5.86 g/L (median of 4.74 g/L), whereas the potassium (K) was between 1.00 and 1.43 g/L (median of 1.22 g/L).

In addition, the concentration of lactic acid was under the limit of quantitation in all wines (<0.50 g/L), confirming that the malolactic fermentation did not occur.

### 3.2. Quali-Quantitative Analysis by GC-MS/MS

In Figure 1, Appendix A, the concentration of free and bound aromatic compounds identified in the 17 *Moscato Giallo* grape samples is reported. Specifically, 34 compounds were quantified, including aliphatic alcohols (3), benzenoid compounds (4), monoterpenes (20), norisoprenoids (5), phenols (1) and vanillins (1).

As shown in Figure 1, the monoterpene compounds were quantitatively greater compared to the total free forms (90%) and the total bound forms (92%), highlighting their contribution to the aroma of the *Moscato Giallo* variety. The analysed samples were characterised by a high level of free linalool, *trans*-pyran linalool oxide, diendiol I and diendiol II, and a low content in the “geraniol group” compounds (*β*-citronellol, nerol, geraniol and geranic acid), as reported by Versini et al. [30].

Linalool showed the highest concentration in all samples, even if some studies have reported that diendiol I was the major monoterpenoid at the mature stage of the *Moscato Giallo* grape [31]. This inconsistency could be attributed to the different environmental conditions (light, temperature, etc.) that affect the concentration of the monoterpenoids in grape berries, as demonstrated specifically for linalool and diendiol I by Fenoll et al. [20].

Among the bound form, the monoterpenes that mainly characterise the *Moscato Giallo* grapes were nerol, geraniol, geranic acid, diendiol I and *cis*/*trans*-8-hydroxy linalool in agreement with D’Onofrio et al. [32]. In contrast to the other monoterpenoids, bound diendiol I was found at high concentrations around veraison and decreased during ripening, as reported by Park et al. [33].

From the quantitative point of view, the *Moscato Giallo* grape samples were characterised by a high content of total linalool (between 9 and 37% of total monoterpenes), *trans*-8-hydroxy linalool (from 8 to 13%), *cis*-8-hydroxy linalool (from 3 to 12%), diendiol I (from 7 to 19%), diendiol II (from 4 to 14%) and geraniol (from 5 to 31%). The second most abundant class of molecules in the bound fraction was norisoprenoids. Although they were not present at high concentrations, norisoprenoids play an important role in the formation of the wine-aging bouquet due to their low odour threshold [34].

1-hexanol, *trans*-3-hexen-1-ol and *cis*-3-hexen-1-ol are the principal aliphatic alcohols of *Vitis vinifera* varieties. As reported in Figure 1, these three compounds were present almost entirely in the free form, contributing significantly to the herbaceous/green odour of grape juice [35,36].

The molecules belonging to the other chemical classes (benzenoid compounds, phenols and vanillins) were found mainly in the bound forms (Figure 1), as described by D’Onofrio et al. [32].

In Figure 2, Appendix A, the concentration of free and bound aromatic compounds identified in the 17 *Moscato Giallo* wines is reported. The wine samples were characterised from the aromatic point of view, considering the same VOCs quantified in the grape berries, except for 2-phenylethanol and benzyl alcohol. These compounds, resulting from grapes, are also produced from the metabolism of yeast during the fermentation process [37,38].

Comparing the data reported in Figure 1 and Figure 2, a similar volatile profile between *Moscato Giallo* grape and wine was found. As shown in Figure 2, linalool was the main monoterpenoid in wine samples, with a concentration of free form ranging from 280 to 1800 µg/L (median of 1150 µg/L) and a concentration of bound form between 60 and 220 µg/L (median of 120 µg/L), in agreement with the data reported by Nicolini et al. [39]. Linalool is one of the most important aroma-active compounds among the monoterpenoids, with an odour threshold of about 25 µg/L. Linalool with geraniol, nerol, citronellol, and *α*-terpineol gives floral, fruity, and citrus nuances. 

Diendiol I was the second most abundant monoterpenoid with a concentration ranging from 230 to 950 µg/L (median of 510 µg/L) in free form and from 11 to 54 µg/L (median of 30 µg/L) in bound form. Even if diendiol I is not considered to contribute directly to *Moscato Giallo* aroma due to its low sensory relevance, it is a precursor of odorant monoterpenoids such as hotrienol in wines [40]. This latter is associated with floral, green, and woody notes.

In accordance with Nicolini et al. [39], the concentrations of pyranoid linalool oxide forms were higher than furanoid ones and the *trans* oxide derivatives were higher with respect to the *cis* ones. Moreover, diendiol I was always predominant with respect to diendiol II, as well as for *trans*-8-hydroxy linalool with respect to the *cis* form.

The two profiles differ for the higher concentrations in the free form of diendiol I, *cis/trans*-8-hydroxy linalool, 1-hexanol, and all norisoprenoids in wine. 1-hexanol belongs to the so-called C_6_-compounds, which are formed during pre-fermentative steps, including harvesting, transport, crushing and pressing, as well as during the eventual must heating or grape maceration. This explains the higher amount of this compound in wine than the total amount in grapes [41].

Aroma compounds quantified in *Moscato Giallo* wines, and their olfactory threshold and odour descriptions are detailed in Table 3.

According to the quali-quantitative results of the volatile fraction, the main monoterpenes (geraniol, linalool, diendiol I, diendiol II, geranic acid, *trans*-8-hydroxy linalool, and *cis*-8-hydroxy linalool) characteristic of *Moscato Giallo* grapes and wine were further investigated by means of isotopic analysis. For the stable isotope ratio analysis, free and bound VOCs were merged, as reported above. 

To avoid an isotopic fractionation of VOCs during the crushing step, a quantitative transfer of the monoterpenes from grapes to juice should be desirable. In Figure 3 reports the correlation between the concentration of the seven main monoterpenes in grape samples and the concentration of the same compounds in the corresponding wines. The concentration of monoterpenes in grapes was corrected considering an average wine yield of 67% after berries processing.

As shown in Figure 3, a good linear correlation for all seven compounds was obtained with a mean correlation coefficient (R^2^) equal to 0.98.

### 3.3. Evaluation of the Isotopic Fractionation along the Concentration Process 

Given the higher detection limit of the IRMS compared to other mass spectrometers, high concentrations of analyte are required for the isotopic determination. For this reason, the samples were concentrated by solvent evaporation, as previously reported in Section 2. As described by Khatri et al. [25], the minimum concentration required to obtain a proper signal for IRMS detection is 25 and 50 mg/L for carbon and hydrogen, respectively.

To evaluate a possible fractionation effect due to the concentration process, an azeotropic mixture (dichloromethane/pentane, 1:2) containing the three target monoterpenoids (eucalyptol, linalool and geraniol) was evaporated in a bath at 40 °C until it reached a final volume of 0.5 mL.

Eucalyptol was chosen for its relatively low boiling temperature, whereas linalool and geraniol were investigated since they are the main monoterpenes in the *Moscato Giallo* variety.

To monitor the isotopic values during the concentration process, aliquot samples were collected every 10 min, and the δ^13^C and δ^2^H values of each of the three compounds were measured by GC-C/Py-IRMS. As shown in Figure 4 and Figure 5, no relevant isotopic fractionation was found in either carbon or hydrogen (below 0.9‰ and 1.9‰, respectively).

### 3.4. Isotopic Values of Grape Must and Wine from Moscato Giallo

As reported in the literature [59,60], yeast metabolism does not affect the monoterpenes during alcoholic fermentation. This claim was also demonstrated from the isotopic point of view, as no significant differences were observed in the δ^13^C and δ^2^H values of the considered monoterpenes before and after the fermentation process (Figure 6).

The mean δ^13^C value measured for the target monoterpenes was relatively depleted if compared with the data reported in the literature for other vegetal matrix [61,62], especially for linalool (−36.6‰) and diendiol II (−35.9‰). This may be due to the type of climate characterising the alpine regions of Italy; usually, plants grown in cold and wet climates show more depleted δ^13^C values [63].

Additionally, Spangenberg et al. [22] reported that the δ^13^C values are generally higher with an increasing vine water deficit, which is not the presented case as all the samples come from regions rich in water (1200 mm/year). However, these unusual negative isotopic data were immediately explained by considering monoterpenes biosynthesis in grapes. In a previous research, Luan et al. [64] demonstrated how monoterpenes biosynthesis in grapes occurs by means of the mevalonate-independent 1-deoxy-d-xylulose-5-phosphate/2C-methyl-d-erythritol-4-phosphate (DOXP/MEP) pathway in plastids. However, this metabolic way is not considered to be the preferred pathway for isoprenoid biosynthesis, which is typically performed by means of the mevalonate acid pathway. Moreover, the differences in these two metabolic pathways reflect a different isotopic fractionation for the secondary metabolites [65]. Dealing with carbon isotopes, Hayes [65] described how a higher depletion of δ^13^C for isoprenoids is observed in the MEP pathway with respect to the same ones synthesised by the mevalonate acid pathway. These findings were able to explain such negative isotopic values registered for monoterpenes in grape berries. In agreement, samples with target volatiles having more negative δ^13^C values with respect to typical C3 distribution may be very useful for authenticity assessment [62,66]. Dealing with fruit aromas, Strojnik et al. [66] provided a clear differentiation for VOCs in apple aromas, since synthetic standards from coal showed much more positive δ^13^C values with respect to natural samples. In this regard, future studies will deal with the analysis of commercial oenological products for authenticity aims. 

Few studies are presented in the literature that report the hydrogen (δ^2^H) isotopic values of VOCs. The trend of the δ^2^H values obtained for analysed VOCs was different from the δ^13^C one (Figure 6). The highest δ^2^H values were always related to diendiol II, and *cis/trans*-8-hydroxy linalool (−150‰), whereas geranic acid and diendiol I showed intermediate values around −180‰.

More depleted δ^2^H values were found on average for linalool and geraniol (−230‰), in agreement with the data reported by Cuchet et al. [62]. Khatri et al. [61] reported how linalool is more depleted than geraniol in lavandin and lavender essential oils, while Hanneguelle et al. [67] found δ^2^H values of linalool between −160‰ and 270‰ by analysing a wide variety of plants (lavender, bergamot, geranium and others).

As regards the carbon isotopic composition of the seven VOCs considered for the isotopic characterisation (linalool, geraniol, diendiol I, diendiol II, *trans*-8-hydroxy linalool, *cis*-8-hydroxy linalool, and geranic acid), although different absolute δ^13^C values were found among the samples investigated, a typical relative trend was found among the terpenes in terms of isotopic distribution. These findings were satisfactory since mono-varietal samples were investigated. Due to the high similarity in terms of geographical origin, no relevant differences were found, as expected, among samples from Trentino and Veneto for isotopic data. 

### 3.5. Isotopic and VOCs Combination for Differentiation of Grape Must and Wine from Moscato Giallo

PCA is one of the most useful tools for the classification of samples according to their chemical composition or isotope ratio analysis [68,69,70] and has the advantage of being unsupervised. In the present study, grape must and wine samples seemed not to be separated by PCA based on VOCs and isotopic data. In particular, the VOCs and VOCs + compound-specific isotopic PCAs (Appendix A) show a quite complete overlapping of the grape musts and the corresponding wines meaning that the main aroma compounds characteristics of *Moscato Giallo* are transferred along the winemaking process. The PCA based only on compound specific isotopic data (Appendix A) highlights an overlap of the must and wine group even if not complete as for VOCs.

## 4. Conclusions

In this study, a full characterisation of the monoterpenic profile in *Moscato Giallo* grapes was achieved by means of the coupling of different analytical approaches. Fast GC-MS/MS analysis provided a suitable quali-quantitative investigation of the key volatiles in grapes in reduced times. Dealing with compound-specific isotopic analysis was demonstrated to be effective in the characterisation and verification of the authenticity of declarations of *Moscato Giallo* wines, as these parameters seem to be not affected during the winemaking process. In this regard, the analysis of *Moscato Giallo* grapes revealed a typical behavior for both VOCs’ composition and isotopic ratios. Current efforts were aimed at increasing the database of grapes investigated, to evaluate specific statistical trends, confirming the genuineness range established for natural samples. 

## Figures and Tables

**Figure 1 biomolecules-14-00710-f001:**
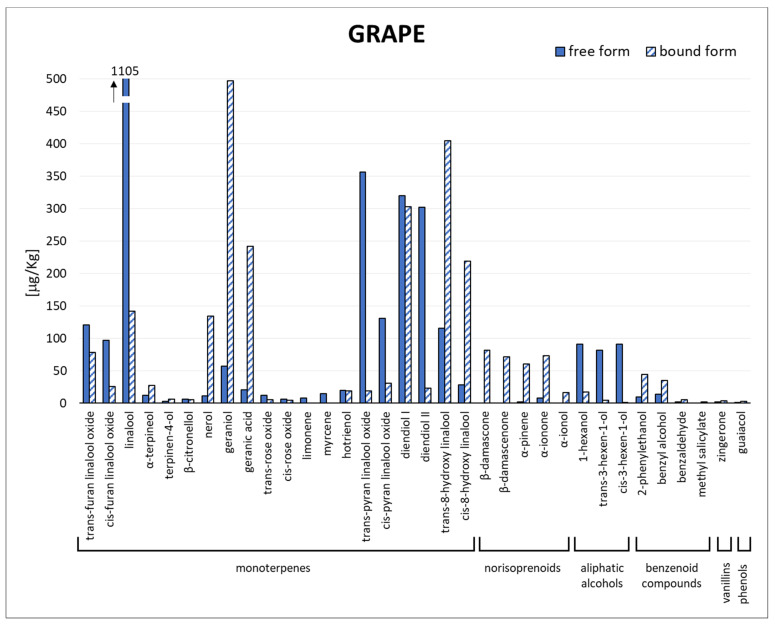
Quali-quantitative profile of free and bound volatile compounds in *Moscato Giallo* grape berries (mean values) collected in Trentino-Alto Adige and Veneto regions.

**Figure 2 biomolecules-14-00710-f002:**
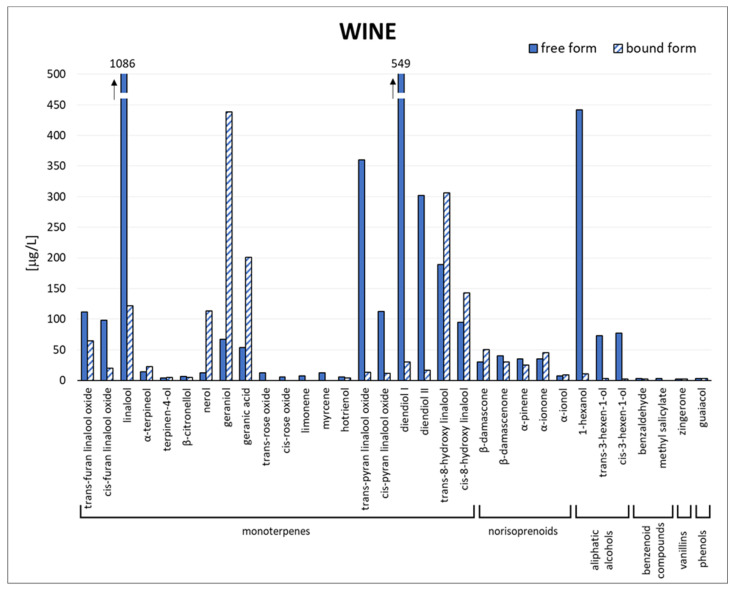
Quali-quantitative profile of free and bound volatile compounds in *Moscato Giallo* wines (mean values) from grapes collected in Trentino-Alto Adige and Veneto regions.

**Figure 3 biomolecules-14-00710-f003:**
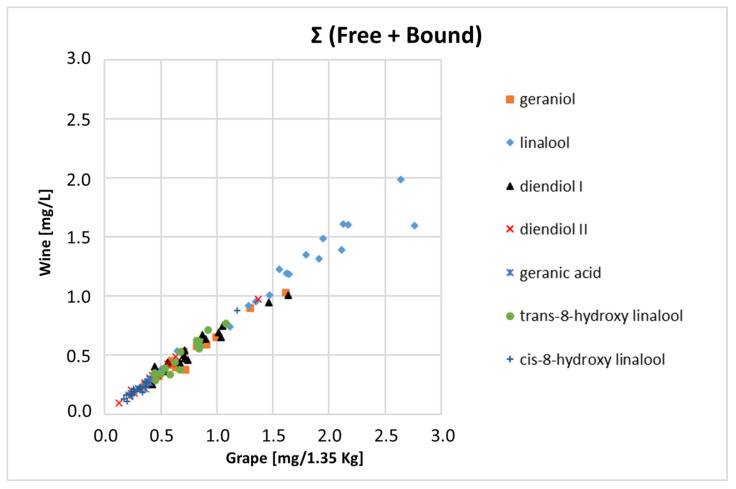
Comparison between the relative concentrations of target monoterpenes quantified in the 17 grape samples and in the corresponding 17 wines of *Moscato Giallo*.

**Figure 4 biomolecules-14-00710-f004:**
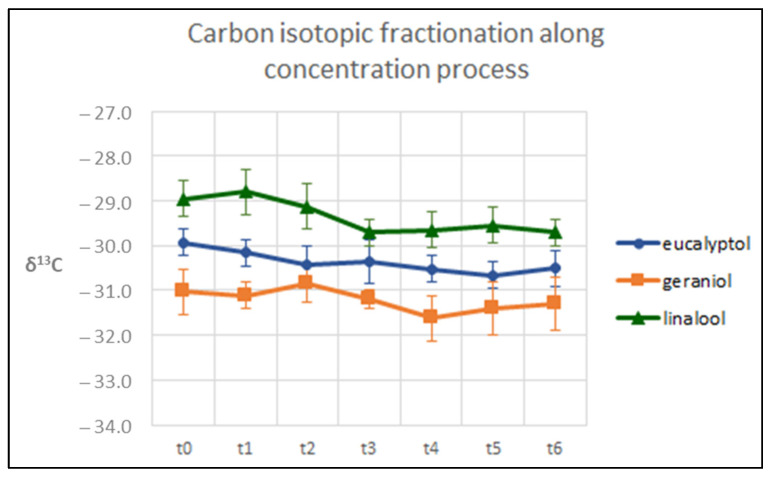
Variation in δ^13^C values during the concentration process for eucalyptol, linalool and geraniol (t0 = starting volume, t6 = final volume).

**Figure 5 biomolecules-14-00710-f005:**
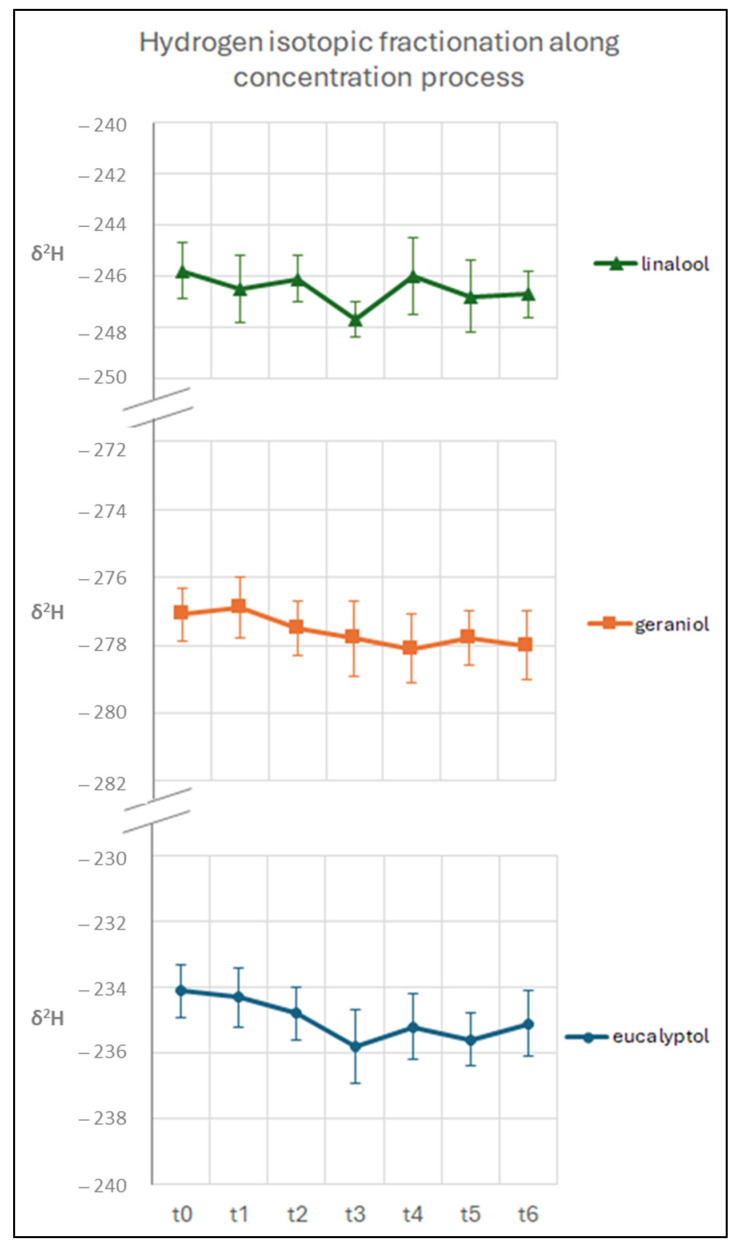
Variation in δ^2^H values along the concentration process for eucalyptol, linalool and geraniol (t0 = starting volume, t6 = final volume).

**Figure 6 biomolecules-14-00710-f006:**
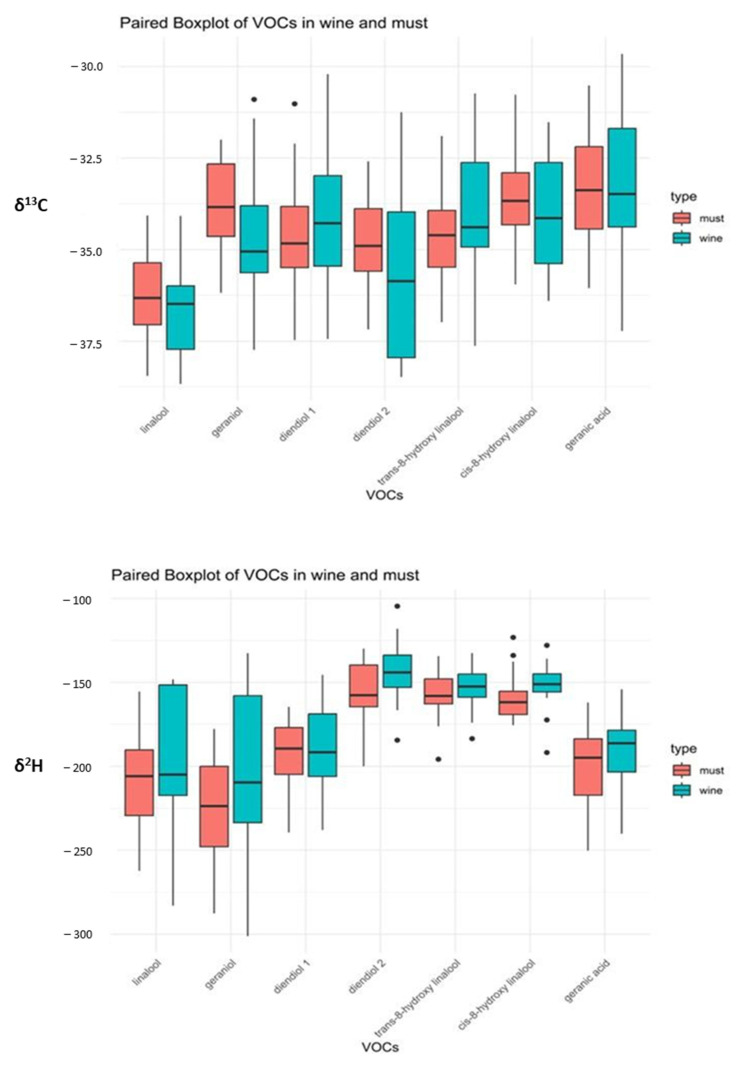
The bar charts of the variability of δ^13^C and δ^2^H values of the seven VOCs considered for the isotopic characterisation of the *Moscato Giallo* grapes and the corresponding wines.

**Table 1 biomolecules-14-00710-t001:** Composition of grape musts. YAN: yeast assimilable nitrogen, α-aa: nitrogen concentration deriving from amino acids; NH_4_^+^: Nitrogen concentration deriving from ammonium.

	BRIX (°)	Reducing Sugars (g/L)	pH	Titratable Acidity (g/L)	Density (g/mL)	Tartaric Acid (g/L)	Malic Acid (g/L)	K (g/L)	YAN (mg/L)	NH_4_^+^ (mg/L)	α-aa (mg/L)
Sample 1	20.03	181	3.22	7.62	1.07718	4.76	5.44	1.36	155	28	127
Sample 2	22.83	172	3.12	7.14	1.07208	4.98	4.48	1.17	<20	<20	<20
Sample 3	21.19	188	3.23	4.81	1.07795	5.01	2.65	1.32	25	<20	23
Sample 4	21.16	152	3.08	7.74	1.06447	4.86	5.10	1.00	83	27	56
Sample 5	20.84	182	3.18	6.76	1.07677	4.76	4.19	1.27	54	<20	46
Sample 6	22.00	185	3.19	6.23	1.07743	4.78	4.20	1.29	58	<20	48
Sample 7	21.76	225	3.35	5.01	1.09410	4.71	2.46	1.53	<20	<20	<20
Sample 8	22.47	189	3.28	4.68	1.07872	4.79	2.46	1.33	69	<20	62
Sample 9	23.55	189	3.29	5.76	1.07925	4.40	3.52	1.37	57	<20	52
Sample 10	21.64	217	3.55	6.39	1.09341	5.47	4.93	2.06	152	<20	135
Sample 11	22.81	201	3.40	5.67	1.08498	5.13	3.29	1.63	81	<20	76
Sample 12	21.93	202	3.32	6.54	1.08573	4.79	4.13	1.56	109	<20	92
Sample 13	20.09	192	3.26	7.02	1.08180	5.42	4.30	1.48	120	23	97
Sample 14	21.02	169	3.26	7.30	1.07228	5.03	4.95	1.39	163	42	120
Sample 15	20.40	193	3.41	6.40	1.08209	3.65	5.45	1.58	137	17	120
Sample 16	20.42	208	3.41	5.15	1.08787	5.39	2.64	1.60	69	<20	69
Sample 17	20.58	182	3.30	7.28	1.07741	4.24	5.51	1.42	140	23	118

**Table 2 biomolecules-14-00710-t002:** Concentration of the main basic quality control parameters of wines at the end of the alcoholic fermentation.

	Ethanol (% *v*/*v*)	Reducing Sugars (g/L)	Non-Reducing Extract (g/L)	pH	Total Acidity (g/L)	Volatile Acidity (g/L)	Density (g/mL)	Tartaric Acid (g/L)	Malic Acid (g/L)	Lactic Acid (g/L)	K (g/L)
Sample 1	11.13	<1.00	24.39	3.35	7.57	<0.10	0.99599	1.63	5.30	<0.50	1.30
Sample 2	11.60	<1.00	25.07	3.08	7.96	<0.10	0.99716	2.16	4.19	<0.50	1.03
Sample 3	11.64	<1.00	23.55	3.36	5.97	<0.10	0.99612	1.11	3.68	<0.50	1.08
Sample 4	11.84	<1.00	25.52	3.19	8.08	<0.10	0.99722	2.04	4.87	<0.50	1.16
Sample 5	12.07	<1.00	24.81	3.25	7.35	<0.10	0.99553	1.33	4.41	<0.50	1.10
Sample 6	12.28	<1.00	24.07	3.17	8.01	<0.10	0.99738	1.88	4.79	<0.50	1.28
Sample 7	11.06	<1.00	23.71	3.30	7.28	<0.10	0.99582	1.66	4.40	<0.50	1.28
Sample 8	11.08	<1.00	24.28	3.21	6.40	<0.10	0.99527	1.84	3.07	<0.50	1.00
Sample 9	11.16	<1.00	24.73	3.26	7.70	<0.10	0.99623	1.45	4.87	<0.50	1.23
Sample 10	11.03	<1.00	24.03	3.46	7.24	<0.10	0.99528	0.94	5.09	<0.50	1.39
Sample 11	11.97	<1.00	23.44	3.28	6.85	<0.10	0.99731	1.87	4.30	<0.50	1.41
Sample 12	11.62	<1.00	24.62	3.20	8.07	<0.10	0.99686	1.95	4.74	<0.50	1.24
Sample 13	11.06	<1.00	22.10	3.31	7.50	<0.10	0.99580	1.35	4.91	<0.50	1.13
Sample 14	11.88	<1.00	22.81	3.21	7.05	<0.10	0.99666	1.62	4.35	<0.50	1.11
Sample 15	11.81	<1.00	22.75	3.33	7.41	<0.10	0.99676	1.69	4.77	<0.50	1.43
Sample 16	11.80	<1.00	23.13	3.37	6.32	<0.10	0.99629	1.12	3.46	<0.50	1.18
Sample 17	11.84	<1.00	24.94	3.19	8.56	<0.10	0.99755	1.78	5.86	<0.50	1.22

**Table 3 biomolecules-14-00710-t003:** Odour threshold and sensory descriptors of odour-active compounds quantified in *Moscato Giallo* wines (n.d. = not defined).

Chemical Class	Compound	Odour Threshold [μg/L]	Sensory Descriptors
Monoterpenes	*trans*-furan linalool oxide	3000 [42]	Sweet, floral [43]
	*cis*-furan linalool oxide	6000 [42]	Floral, sweet, woody [43]
	linalool	25 [44]	Citrus, floral, sweet [45]
	*α*-terpineol	250 [42]	Floral, sweet [45]
	terpinen-4-ol	250 [45]	Sweet, herbaceous [45]
	*β*-citronellol	100 [46]	Lemongrass [46]
	nerol	400 [46]	Lime, floral-hyacinth, roses [46]
	geraniol	30 [44]	Rose, geranium [44]
	geranic acid	40 [47]	Green [47]
	*trans*-rose oxide	100 [44]	Rose-like, floral, sweet [43]
	*cis*-rose oxide	100 [44]	Floral, lychee-like, rose [44]
	limonene	200 [48]	Orange, mint, lemon, floral, citrus [49]
	myrcene	14 [50]	Green, floral, grass, citrus [49]
	hotrienol	110 [43]	Floral, green, woody [43]
	*trans*-pyran linalool oxide	3000–5000 [51]	Sweet, floral, earthy [43]
	*cis*-pyran linalool oxide	3000–5000 [51]	Sweet, floral, earthy [43]
	diendiol I	n.d.	n.d.
	diendiol II	n.d.	n.d.
	*trans*-8-hydroxy linalool	n.d.	Floral [43]
	*cis*-8-hydroxy linalool	n.d.	Floral [43]
Norisoprenoids	β-damascone	0.09 [44]	Fruity-flowery, exotic-spicy [52]
	β-damascenone	0.05 [44]	Apple, rose, honey [44]
	*α*-pinene	n.d.	Woody, resinous [53]
	*α*-ionone	2.6 [54]	Violets, berry [55]
	*α*-ionol	n.d.	Floral and woody [55]
Aliphatic alcohols	1-hexanol	8000 [44]	Resin, green (cut grass) [44]
	*trans*-3-hexen-1-ol	1000 [54]	Grassy green, earthy [56]
	*cis*-3-hexen-1-ol	400 [44]	Lettuce-like, green, grass [44]
Benzenoid	benzaldehyde	2000 [57]	Bitter almond [57]
	methyl salicylate	50 [58]	Mint-like [43]
Vanillins	zingerone	n.d.	Toasty, dry fruit
Phenols	guaiacol	10 [44]	Smoke, sweet, medicine [44]

## Data Availability

Not applicable.

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
