# Peer review of "Aromatic Characterisation of *Moscato Giallo* by GC-MS/MS and Validation of Stable Isotopic Ratio Analysis of the Major Volatile Compounds"

_biomolecules, 2024, doi:10.3390/biom14060710_

Round 1
Reviewer 1 Report
Comments and Suggestions for Authors
Please see attachments.

English is mostly fine.
Author Response
Reviewer 1
General comments. This is a paper describing relationships between grape and wine volatile compound concentrations in Moscato Giallo, and, their relationships with carbon (δ13C) and hydrogen (δ2H) isotopic signatures of the major volatile compounds. It is generally a well-written paper. Figures and tables are mostly legible and well prepared; results are adequately described and discussed. It can be accepted after moderate revisions that include combination of some figures and either deletion or moving other figures to Supplemental. I have listed the issues below. I also have provided an edited version of the pdf that might assist the authors.
We thank the reviewer for the valuable comments and suggestions, below we describe all the modifications done point by point according to the reviewer’s suggestions.
Abstract 20; elsewhere
Normally chemical names are not uppercase. I suggest make all lowercase throughout the manuscript.
The chemical names were corrected as suggested by the reviewer.
Introduction 37-88 I suggest breaking this first paragraph into 2-3 paragraphs or more. At the moment, it is a "wall of words". I particularly highly recommend making the information regarding importance of δ13C and δ2H front and center by breaking this into separate paragraphs. One break could be placed in line 51 (For detecting fraudulent…) and another in line 63 (Oxygen and hydrogen…).
The text was modified according to the suggestion.
Materials & Methods
137 Source of yeast? Lallemand?
There was a typo in the text, we corrected it specifying that the used yeast was a laboratory yeast strain and not a commercial one.
144 Suggest use of "variables" and not "parameters" here and elsewhere
In this specific case we think that it is better to keep the term parameters as they are used to define and characterise the products we are analysing.
192-193 Check end of line hyphenation for “Mas- sHunter Workstation”
Corrected.
204 Normally city and US state are listed for manufacturer.
The information was added.
Results & Discussion
246 Table 1: This table needs to be expanded horizontally--the column titles and some data do not fit the columns. It may require hyphenating some column titles, reducing the font size, deleting one or more columns, etc. Also, make sure this and all other tables and figures stand on their own— i.e. location of samples, etc. What is "density" and why are there two numbers in each cell?
The Table and the captions were modified according to the reviewer’s suggestions. As regards the density (i.e. mass/volume), there was a formatting problem and we solved it.
266 As with Table 1, this table needs to be expanded horizontally--the column titles and some data do not fit the columns. Perhaps the reducing sugar and lactic acid columns can be deleted.
Corrected as suggested
311 For this and other figures, please revise the figure captions to ensure that each figure stands on its own. Add information such as source of grapes/wines, etc. Same goes with tables.
The requested Information were added and the captions revised
380-392 Figures 4 and 5 are each three-window figures that have a lot of "white space" and take up a lot of room. Could they be reduced in size to one 6- window composite figure that occupies one page or less? Also, these figures represent only three compounds. What about all the others? If all the other terpenes and other compounds show similar results, why include three compounds? Why not just one representative compound for grapes and wines?
Figures 4 and 5 were modified as per reviewer’s suggestion. We included in this validation part to check a possible fractionation effect due to the concentration process the three terpenes eucalyptol, linalool and geraniol because, as detailed in the text, eucalyptol for its relatively low boiling temperature, linalool and geraniol as they are the main monoterpenes in Moscato Giallo variety.
423-428 Figure 6: Please define "VOCs" or simply spell out.
The definition of VOCs is included in line 104 in the Introduction part.
457-462 Figure 7: I'm not sure these figures provide much insight into the data nor add overall to the message of the paper. Consider placing these in the 'Supplemental' section. Also, as in Figure 6, please "VOCs" or simply spell out.
As suggested by the reviewer figure 7 was moved in the “Supplementary Material section”
References - Please ensure that all titles of journal papers are in lowercase except proper nouns. See reference # 13, 17, 44; maybe others.
Corrected

Reviewer 2 Report
Comments and Suggestions for Authors
This is an interesting manuscript that analyzed the components in our daily life product. Authors used fast GC MS analysis to quali-quantitative investigated the volatiles organic compounds in grapes wines. Not many studies have done this topic research, but several analyses of similar organic compounds can be found in literature. I suggest authors consider more scientific meaning rather than just data presents.
Major comments: Authors should consider one paragraph illustrate the relationship between these analyzed compounds and their ratios to the wine/grape taste or smile. This will provide more general audience scientific meaning of this research results. The isotope ratios usually reflect the geographical features, if this can relate to the location, weather, or even soli condition would be great.
Minor comments:
1. There are no typical GC MS figures shown in this manuscript, authors could consider not to address that in the title.
2. Figure 3 can adjust the y-axis to make it clear.
3. Possible to make the figures prettier instead of very clear excel format?
4. Figure 4,5 are three different size figures, authors can consider combining into one like Figure 3.
Author Response
Reviewer 2
Comments and Suggestions for Authors
This is an interesting manuscript that analyzed the components in our daily life product. Authors used fast GC MS analysis to quali-quantitative investigated the volatiles organic compounds in grapes wines. Not many studies have done this topic research, but several analyses of similar organic compounds can be found in literature. I suggest authors consider more scientific meaning rather than just data presents.
Major comments: Authors should consider one paragraph illustrate the relationship between these analyzed compounds and their ratios to the wine/grape taste or smile. This will provide more general audience scientific meaning of this research results.
As suggested by the reviewer we added a Table (Table 3) with all the information related to the VOCs and their relationship with wine taste and flavour including their threshold limits of perception.
The isotope ratios usually reflect the geographical features, if this can relate to the location, weather, or even soli condition would be great.
Unfortunately, as the focus of the study was to verify the feasibility of a combined GS-MS-MS and GC-IRMS approach to improve the characterization of Moscato wines, the samples were collected from a very narrow area without any remarkable differences according to the geo-climatic characteristics. Therefore, the isotopic ratios in this specific case were not significantly different according to the geographical origin.
Minor comments:
- There are no typical GC MS figures shown in this manuscript, authors could consider not to address that in the title.
As the aim of the study was to develop a combined approach GC-MS-MS and GC-IRMS we prefer to keep the reference to GC-MS analysis in the title. Figures related to GC-MS-MS are reported in Figures 1 and 2 and we added 2 new Tables with all the data per sample (S1 and S2)
- Figure 3 can adjust the y-axis to make it clear. We modified the Figure as suggested.
- Possible to make the figures prettier instead of very clear excel format? We tried to modify the figures according to the reviewer's comment.
- Figure 4,5 are three different size figures, authors can consider combining into one like Figure3. Figures 4 and 5 were combined as per reviewer’s suggestion.